# The Application of Adaptive Model Predictive Control for Fed-Batch *Escherichia coli* BL21 (DE3) Cultivation and Biosynthesis of Recombinant Proteins

Konstantins Dubencovs [1,2] , Arturs Suleiko [1], Elina Sile [3] , Ivars Petrovskis [4], Inara Akopjana [4], Anastasija Suleiko [1], Vytautas Galvanauskas [5] , Kaspars Tars [4] and Juris Vanags [1,2,*]

1    Laboratory of Bioengineering, Latvian State Institute of Wood Chemistry, Dzerbenes Street 27, LV-1006 Riga, Latvia; gmtd@inbox.lv (K.D.); arturs.suleiko@bioreactors.net (A.S.); anastasija.gurcinska@gmail.com (A.S.)
2    Institute of General Chemical Engineering, Faculty of Materials Science and Applied Chemistry, Riga Technical University, Paula Valdena Street 3, LV-1048 Riga, Latvia
3    Institute of Applied Chemistry, Faculty of Materials Science and Applied Chemistry, Riga Technical University, Paula Valdena Street 3, LV-1048 Riga, Latvia; elina.sile@rtu.lv
4    Latvian Biomedical Research and Study Centre, Ratsupites Street 1, LV-1067 Riga, Latvia; vars@biomed.lu.lv (I.P.); inara@biomed.lu.lv (I.A.); kaspars@biomed.lu.lv (K.T.)
5    Department of Automation, Kaunas University of Technology, LT-51367 Kaunas, Lithuania; vytautas.galvanauskas@ktu.lt
*    Correspondence: juris.vanags@bioreactors.net

**Abstract:** A model predictive control (MPC) method was investigated as a route to optimize and control the growth of *E. coli* BL21 (DE3) and biosynthesis of two different recombinant proteins (nerve growth factor NGF and coat protein of bacteriophage Qβ (Qβ-CP)). To determine the target trajectory for the *E. coli* cultivation process and estimate the model parameters, the off-line run-to-run optimization method was used. The proven method allowed us to successfully control the growth of microbial biomass, with a deviation of 6–12% from the target trajectory. It was proven that it is possible to obtain a "Golden Batch" profile for the implementation of MPC using datasets from only four to eight fermentation runs. The method showed its robustness when the cultivation of *E. coli* was carried out with two different titrant supply control systems—volumetric and gravimetric. Furthermore, the MPC method exhibited high adaptability, reliability, and resistance to various types of disturbances. MPC proved to be a reliable and effective method for controlling the cultivation and recombinant protein biosynthesis of fast-growing microorganisms such as *E. coli*.

**Keywords:** model predictive control (MPC); fed-batch fermentation; *E. coli*; recombinant protein

## 1. Introduction

Recent progress in the field of gene transformation has led to an increase in the demand for effective cultivation control systems. This would accelerate the research on new high-potential microorganism strains and products, which can be produced on a commercial scale. One of the main cultivation technology development challenges is the variability of microorganism growth trajectories, which can take place between multiple batches, even if the microorganism is kept under the same conditions. For example, Jenzsch et al. demonstrated rather low repeatability in terms of yield during *Escherichia coli* (*E. coli*) cultivations when recombinant proteins were the target product. The variability in biomass growth trajectory and recombinant protein yield between 13 batches, which were subjected to identical conditions, was about 20% and 30%, respectively [1].

Traditionally, prior to fermentation processes, time-dependent feeding profiles are adjusted. During the process, the substrate is fed according to this pre-determined profile, and an operator manually readjusts the profile based on the results of off-line substrate analysis.

However, due to the inherent variability in microorganism growth, these traditional process control methods may not always guarantee high reproducibility during cultivations [2]. Especially, the mentioned problem often occurs during fed-batch fermentation while selecting the optimal feeding rate profile, which would facilitate biomass growth along a desired trajectory and yield high product titers [1]. By applying adjusted substrate profile control approaches, it is difficult to maintain the substrate concentration at acceptable values during cultivation. These deviations of the substrate concentration even for relatively short periods of time can cause irreversible changes in the physiology of microorganism cells and negatively impact both the target product yield and quality. Furthermore, among small- to mid-scale production facilities, which operate with cultivation processes, there still exists the trend of implementing direct corrections to the feeding rate profile and/or other process variables by the operating staff, e.g., by hand. The mentioned procedure contradicts the FDA's Process Analytical Technology (PAT) initiative, which states that it is important to ensure automatic real-time process control and minimize human influence on the process [3].

Numerous research groups have dedicated their work towards the development of automatic fed-batch cultivation/fermentation process control systems. As of now, two basic approaches for biological process control are applied: open-loop and closed-loop control. In open-loop control, the feeding rate is predefined by a time vs. feeding rate profile, which is independent from the output parameters of the process, e.g., even if deviations from the target biomass growth trajectory are detected, no control actions (feeding profile corrections) are implemented. On the one hand, the mentioned approach is much more convenient in terms of execution and implementation; on the other hand, due to a lack of feedback, the reproducibility and reliability of processes operated under open-loop control are rather low, especially if the biomass-specific growth rate is close to its maximum [4].

Generally, fed-batch fermentations operated under closed-loop control have better performances in comparison to open-loop. Closed-loop systems use feedback to adjust the control over the fed-batch bioprocess. The input, feedback, and output are continuously monitored and compared, and the updates to the control output are implemented within the set intervals; thus, significantly increased biomass and or product yield can be reached in the process [5,6]. For the implementation of closed-loop control, multiple methods currently are widely applied, such as proportional–integral–derivative (PID) control, probing control, artificial neural networks (ANN), fuzzy control, statistical process control, etc.

At present, the most common closed-loop control method is the PID controller. In the case of PID control, a signal is generated when certain deviations from a predefined target process value are detected. The mentioned signal is then sent to the control element (pump, valve, etc.), which is directly responsible for implementing the required actions, in order to maintain the target value in an optimum range. The value of the control signal is usually dependent on three terms: the first of which is proportional to the difference between the target value and the currently observed value, the second term is proportional to the integral of the error signal, and the third term is proportional to the derivative of the error signal. PID control in fed-batch processes is usually implemented in the form of indirect feedback control schemes that couple the nutrient feeding rate with measurements of pH (pH-stat) and/or DO (DO-stat) [7,8]. The pH-stat is based on the phenomena that pH rises due to excretion of ammonium ions when the carbon substrate is depleted [7]. Similarly, the DO-stat is based on the fact that DO increases sharply when a key substrate is depleted [8]. Other feedback control schemes apart from pH-stat and DO-stat are also widely used in the industry. For example, during the cultivation of baker's yeast, the ethanol concentration in the broth may be used as the controlled parameter, whose concentration is maintained constant. In this way, the biomass-specific growth rate is kept at a maximum while avoiding ethanol formation [9]. PID control shows favorable results as the application complexity is less pronounced and the overall performance of the process is better than in the case of open-loop control [7–9]. Despite the upper mentioned fact, the use of PID control is limited in some cases due to the nonlinearity and time variant nature of biological

processes [10] and the lack of reliable on-line measurement systems for the direct reading of control parameters. For example, it is very difficult to accurately measure parameters such as the concentration of biomass or substrate on-line. Enzymatic sensors used to measure substrate concentrations often do not have sufficient accuracy and reliability, and the readings of optical or capacitive biomass sensors are influenced by parameters such as mixing, aeration, foam, and the state of a culture. More accurate measurements of the mentioned control parameters can be obtained by in-line chromatography or flow cytometry; nevertheless, the mentioned approaches do not guarantee that the state of the system will be correctly evaluated. For example, the lack of culture growth does not necessarily indicate malnutrition but may be a result of metabolite formation that inhibits biomass growth. Also, maintaining a certain critical substrate concentration ($S_{crit}$) does not guarantee that there is no overflow metabolism in the system since $S_{crit}$ can change from process to process [11].

The probing control method is based on supplying a short pulse control action (short-term change in the feeding rate) and assessing the response of the system (change in DO) in order to subsequently correct the control setpoint [12–15]. The method is itself adaptable, quite easily implemented in modern PLCs (programmable logic controllers), and has proven to be applicable to cultivation processes of *E. coli*, although the probing control is hardly applicable to higher organisms, systems with high inertia (large volume, low biomass growth rate) [13], and bioprocesses utilizing complex nutrient mediums (multi-substrate) [15].

Artificial neural network (ANN) [16,17], statistical process control [18], and fuzzy logic [19] methods use empirical models based on the analysis of experimental data to form a control action. The mentioned methods show favorable results if the system parameters do not deviate significantly from the parameter values that were used during model creation. The main drawback of these methods is the need to create a model for each new process, which requires the collection of a large amount of statistical data.

As of now, one of the most promising closed-loop control methods is model predictive control (MPC), which for the formation of a control action (feeding rate correction) evaluates the deviations between the predicted and reference values of the controlled variable (biomass or substrate concentration, biomass specific growth rate, etc.). Reference trajectories can be obtained from already developed processes [20], calculated using mathematical models [1] or identified experimentally, for example, by using the probing method [15]. Empirical models (neural models—ANN [16,17], partial least squares [18]) and classical mechanistic mathematical models [2,21] can be used for developing a prediction. Empirical models are quite accurate within the range of process parameters from which they were compiled but are poorly scalable and have less predictive and analytical power compared to mechanistic models [18]. In addition, the presence of the predicting model of the system's behavior, as well as the reference trajectory, allows us to forecast possible deviations of the process in advance, which is not possible using other methods of direct control.

Many examples of MPC applications for the control of yeast [16,17], bacteria [1,21], and mammalian cells [2,20], as well as the biosynthesis of alcohol [16,17], recombinant proteins [20], hormones [1], and antibiotics [18], can be found in the literature. Most of these studies were carried out with various kinds of simulators [16,17] and only a small portion were dedicated towards direct operation in real systems.

The aim of our study was to develop an adaptive MPC-based process control system that would not be technically overly complex, thus ensuring its applicability to a sufficiently larger number of users in the scientific research community and biotechnology-oriented manufacturing. It should be noted that currently MPC-based systems for the control of fermentation/cultivation processes are not yet available on the market and must be individually designed for specific purposes. Thus, our main goal was to develop a system which could be adapted to different kinds of biotechnological applications (for different cell cultures, cultivation strategies, and target products).

The design of a control system based on MPC would give an opportunity to expand the effective use of bioreactors in the implementation of various fed-batch processes. There are several reasons why fed-batch process control cannot be ensured using only a standard PLC. In such cases, the response to impacts has a pronounced delay and mathematical models must be applied to ensure process control. These circumstances determine the perspective of using MPC-based control because such system programs, which are installed on standard PC systems, can be connected to bioprocess controllers. In this way, there are ample opportunities to significantly supplement the capabilities of standard PLCs. Among the above-mentioned approaches in fed-batch control, MPC is practically the only method that can provide the use of mathematical models in the control of biotechnological processes using standard PC equipment.

In this study, we utilized a nonlinear adaptive MPC algorithm that we developed. The algorithm is based on a simplified unstructured mechanistic model. Here, we demonstrate the effectiveness of this algorithm in both the development and control of the fed-batch cultivation process of *Escherichia coli* BL21 (DE3) and the biosynthesis of two different recombinant proteins: nerve growth factor NGF and coat protein of bacteriophage Qβ (Qβ-CP), able to self-assemble in virus-like particles (Qβ-VLPs). Both proteins are used in vaccine development in our labs. NGF can be used as a target to attenuate chronic pain, which has been demonstrated in a murine osteoarthritis model [22]. Qβ-VLPs have been used as carriers in many different VLP vaccine candidates against a variety of communicable and non-communicable diseases and conditions including hypertension [23], smoking addiction [24], COVID-19 [25], and Lyme disease [26]. Therefore, both proteins need to be produced in large quantities and can benefit from production optimization methods.

## 2. Materials and Methods

### 2.1. Experimental Procedures

#### 2.1.1. Microorganism and Culture Media

All fermentations were performed with *E. coli* BL21 (DE3) provided by the Latvian Biomedical Research and Study Centre. The nerve growth factor NGF was expressed under the control of the T7 promoter after induction with 1 mM of isopropyl-thiogalactopyranoside (IPTG). The strain was resistant to kanamycin. The product appears as an inclusion body within the cytoplasm. Also, the Qβ CP was expressed under the control of the T7 promoter after induction with 1 mM of IPTG. The strain was resistant to ampicillin.

The culture medium for inoculum preparation and fed-batch cultivations was prepared according to R. Bajpai [27] with no modifications and comprised 5.5 g/L glucose, 5 g/L yeast extract, 3.5 g/L $(NH_4)_2SO_4$, 5.6 g/L $KH_2PO_4$, 0.45 g/L $K_2HPO_4$, 0.35 g/L $MgSO_4 \cdot 7H_2O$, 0.003 g/L $CaCl_2 \cdot 2H_2O$, 0.017 g/L $FeCl_3 \cdot 6H_2O$, 0.0065 g/L $ZnSO_4 \cdot 7H_2O$, 0.006 g/L $Na_2MoO_4 \cdot 2H_2O$, 0.006 g/L $CoCl_2 \cdot 6H_2O$, 0.003 g/L $CuCl_2 \cdot 2H_2O$, 0.003 g/L $H_3BO_3$, 0.01 g/L thiamine. For fed-batch fermentation, a feeding solution was used that consisted of salts and glucose; the concentrations were as follows: 400 g/L glucose, 94 g/L $(NH_4)_2SO_4$, 0–5.6 g/L $KH_2PO_4$, 0–0.45 g/L $K_2HPO_4$, 5.56 g/L $MgSO_4 \cdot 7H_2O$, 1.32 g/L $CaCl_2 \cdot 2H_2O$ (20).

#### 2.1.2. Inoculum Preparation

The inoculum was prepared in 250 mL shake flasks containing 100 mL culture medium. The pH was adjusted to 7.0 by adding 10% (*w/v*) NaOH solution. Each flask was inoculated with 0.2 mL of *E. coli* glycerol stock. The flasks were incubated for 24 h at 37 °C and 220 rpm (orbital shaker PSU-20i, Biosan, Latvia).

#### 2.1.3. Bioreactor Cultivation Studies

The cultivations were performed in a lab-scale bioreactor system EDF-5.4/BIO-4 (Biotehniskais Centrs, Riga, Latvia). EDF-5.4/BIO-4 consisted of a 5.4 L glass reactor with

a working volume of 2–4 L. The vessel was equipped with two standard Rushton turbines and the bioprocess controller—BIO-4.

The cultivations were started in the batch mode with an initial volume of 2 L, after inoculation of 100 mL of inoculum, and proceeded as fed-batch when the MPC software activated feeding (predicted substrate or biomass concentration equal to reference trajectory) to follow predefined reference biomass growth trajectory (after about 4 h).

The pH of the medium was controlled at $7.0 \pm 0.2$ using 30% (*w/v*) sodium hydroxide and 20% (*w/v*) sulfuric acid solutions, and the temperature was kept at $37.0 \pm 0.2$ °C. The oxygen saturation (pO₂) was controlled at 40% by increasing stirrer rotation speed to the allowed maximum (800 rpm), and then enriching the inlet air with pure oxygen. Constant airflow or air/oxygen mixture of 2.0 lpm was maintained during all the processes. The foam level was controlled by adding antifoam A (Sigma). The fermenter overhead pressure was kept at 0.1 bar above ambient.

### 2.2. Analytical Methods

Biomass growth was monitored by off-line measurements of the optical density (*OD*) at 560 nm (Jenway, 6300, Essex, UK). The biomass concentration was calculated by multiplying *OD* with the correlation coefficient of 0.27 determined in advance. Glucose was measured enzymatically (AccuChek ACTIVE, Roche, Basel, Switzerland). Protein production was monitored by SDS-PAGE electrophoresis, followed by staining with Coomasie-blue.

### 2.3. Control System

The cultivaton was carried out applying two different titrant supply control systems: volumetric and gravimetric. A bioreactor with gravimetric control of titrants and an MPC system is shown in Figure 1.

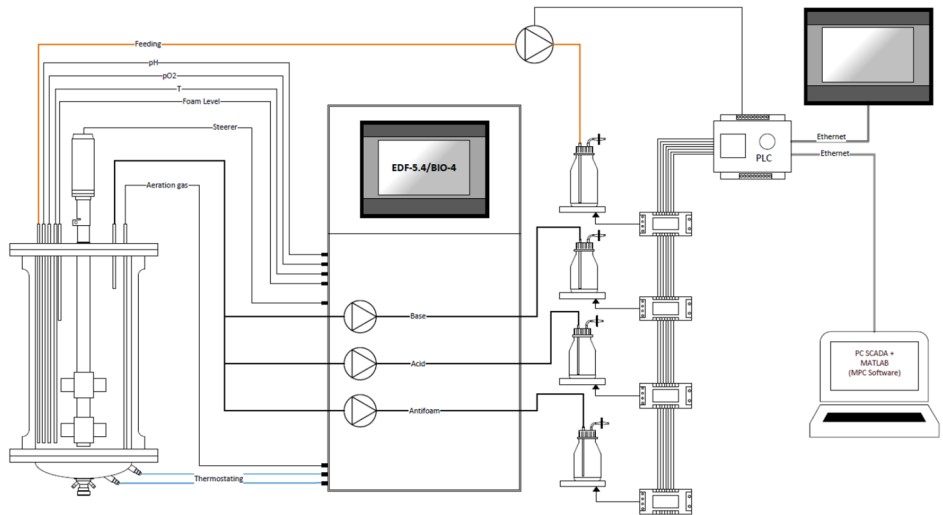

**Figure 1.** The schema of MPC realization.

The main elements of the bioreactor system applied in the current study were (1) an EDF-5.4 laboratory glass bioreactor, (2) a BIO-4 bioprocess controller, (3) a weighing module, and (4) a PC with SCADA (*Supervisory Control and Data Acquisition*) and the MPC software, which was developed in MATLAB. The BTC weighing module was equipped with an analogue peristaltic pump (Longer-Pump, BT100–2J) and four weight platforms. The module was used for feeding rate control and titrant (feeding, acid, base, and antifoam) volume monitoring. Three BCL-3L (CAS, Seul, Republic of Korea) (range 0–3 kg, accuracy 0.0003 kg) and one BC-5A (CAS, Seul, Republic of Korea) (range 0–5 kg, accuracy 0.0003 kg) weighing elements were embedded inside the module for precise weight monitoring for acid, base, antifoam, and feeding solutions, respectively. The glass flasks, which contained the titrant solutions (acid, base, antifoam, and feeding) were each placed on its individual

weighing platform. Using either the peristaltic pumps, which were built into the BIO-4 bioprocess controller (in the case of acid, base, and antifoam solutions), and the analogue peristaltic pump, which was connected to the weighing module, solutions were supplied to the bioreactor in accordance with the applied control algorithm (pH control, foam level control, and MPC). The SCADA software was built using the PcVue (PcVue Solutions, ARC Informatique, Paris, France) framework and was used for data exchange between the weighing module, the MPC software, and the bioreactor system. The volumetric titrant supply control system has a similar design, except that the measurement of the amount of substrate supplied is carried out using pre-calibrated peristaltic pumps. In this case, MPC software receives the data and controls the feeding profile directly through the BIO-4 bioprocess controller.

### 2.4. Model Predictive Control Realization

During cultivations, the operator performed off-line glucose and biomass measurements and inputed the results into SCADA, which passed the updated information to the MPC software (see Figure 2).

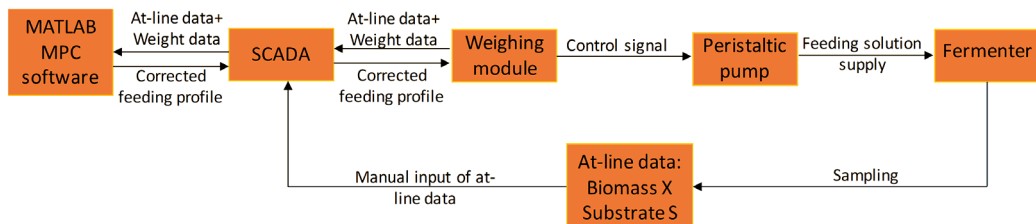

**Figure 2.** Communication diagram of the control system [28].

The MPC software based on off-line data, also considering the deviation of predicted pumped volumes of titrants—acid, base, and antifoam, made a prediction regarding the state of the system within a predefined timeframe (termed the prediction horizon). When a forecast was built, the software compared these predicted trajectories to the reference trajectory and if deviations were observed, the MPC algorithm performed a correction of the feeding rate profile within a predefined timeframe (termed the control horizon), e.g., until new sets of off-line data were supplied to the system. The corrected feeding profile was then uploaded to the PLC of the BTC weighing module, and the peristaltic analogue pump proceeded the operation with the newly updated setpoint values. The on-line data exchange between the MPC software and SCADA, and between SCADA and the PLC of the BTC weighing module, was implemented every second through an OPC server.

### 2.5. Process Model

The mathematical model for process simulation and reference trajectory calculation was based on the model of Levisauskas et al. [29]. Since it was assumed that no limitations of oxygen mass transfer took place and no acetate formation occurred during the cultivation processes, to simplify the model, oxygen consumption and acetate formation were neglected. For the current application, the following differential mass balance equations (Equations (1)–(4)) for biomass, protein, substrate, and volume modeling in the fed-batch process were used:

$$\frac{dX}{dt} = \mu_s X - \frac{F}{V} X \tag{1}$$

$$\frac{dS}{dt} = -\sigma X - \frac{F}{V} S + \frac{F_S}{V} S_f \tag{2}$$

$$\frac{dV}{dt} = F - F_{smp} \tag{3}$$

where X, S, P are the concentrations ($g \cdot L^{-1}$) of biomass, substrate, and target product, respectively, V is the volume of culture broth, L, $S_f$ is substrate concentration in feed, $g \cdot L^{-1}$, μ are the specific biomass growth rate, $g \cdot g^{-1} \cdot h^{-1}$:

$$\begin{aligned} \mu &= \sigma \cdot (Y_{xs} + Y_{xye}) \\ &\text{if } t > t_e \\ \mu &= \sigma \cdot Y_{xs} \end{aligned} \tag{4}$$

$Y_{xs}$—the biomass yield from substrate (glucose), $g \cdot g^{-1}$, $Y_{xye}$—the biomass yield from yeast extract, $g \cdot g^{-1}$, $t_e$—duration of biomass growth on yeast extract, h, σ—the specific glucose consumption rate, $g \cdot g^{-1} \cdot h^{-1}$:

$$\begin{aligned} \sigma &= \sigma_{max} \frac{S}{K_s + S} \cdot \frac{K_{si}}{K_{si} + S} \cdot \left(1 - \frac{X}{K_{Xmax}}\right) \\ &\text{if } t > t_{ind} \\ \sigma &= \sigma_{max\_p} \frac{S}{K_{s\_p} + S} \cdot \frac{K_{si\_p}}{K_{si\_p} + S} \cdot \left(1 - \frac{X}{K_{Xmax\_p}}\right) \end{aligned} \tag{5}$$

$K_s$, $K_{si}$, $K_{Xmax}$, $K_{s\_p}$, $K_{si\_p}$, $K_{Xmax\_p}$—the substrate half saturation, substrate inhibition, and biomass inhibition constants during biomass growth and protein biosynthesis stages, $g \cdot L^{-1}$; $\sigma_{max}$, $\sigma_{max\_p}$—the maximum specific glucose consumption rate during biomass growth and protein biosynthesis stages, $g \cdot g^{-1} \cdot h^{-1}$:

$F_b$ is the base addition rate ($L \cdot h^{-1}$) used during the pH control. It is considered to be proportional to the biomass specific growth rate:

$$F_b = Y_{Xb}^{-1} \cdot \mu \cdot X \cdot V \tag{6}$$

$Y_{Xb}$ is the biomass yield from base, $L \cdot g^{-1}$,

$F_{af}$ is the antifoam addition rate ($L \cdot h^{-1}$) used during the foam control and is proportional to the biomass specific growth rate:

$$\begin{aligned} F_{af} &= Y_{Xaf}^{-1} \cdot \mu \cdot (X - X_{crit}) \cdot V \text{ if } X < X_{crit} \\ F_{af} &= 0 \end{aligned} \tag{7}$$

$Y_{Xaf}$—the biomass yield from antifoam, $L \cdot g^{-1}$; $X_{crit}$—the minimum concentration of biomass, $g \cdot L^{-1}$, when the supply of antifoam agent is required.

$F_c$ is carbon loss rate ($g\ L^{-1} \cdot h^{-1}$) during the respiration:

$$F_c = \frac{(M_{CO_2} - M_{O_2})}{M_{O_2}} \cdot OUR \cdot V \tag{8}$$

$M_{CO_2}$, $M_{O_2}$—molar masses of $CO_2$ and $O_2$, respectively, $g \cdot Mol^{-1}$.

OUR is the oxygen uptake rate:

$$OUR = Y_{OX} \cdot \mu \cdot X + O_m \cdot X \tag{9}$$

$Y_{OX}$—is the biomass yield from oxygen, $g \cdot g^{-1}$, $O_m$—oxygen maintenance coefficient, $g \cdot g^{-1} \cdot h^{-1}$,

F is the rate of the culture volume change, $L \cdot h^{-1}$:

$$F = F_S + F_b + F_{af} - F_c - F_e \tag{10}$$

$F_e$ is the evaporation rate, $F_s$—the substrate addition rate, $L \cdot h^{-1}$:

$$\begin{aligned} F_s &= f(t) \\ F_S &= (\sigma \cdot X \cdot V + (F_b + F_{af} - F_c - F_e) \cdot S) / (S_f - S) \end{aligned} \tag{11}$$

Process modeling was performed in MATLAB environment. For the integration of differential equations, the ODE solver ode15s (a variable-order method for solving stiff differential equation systems) was used.

Reference trajectories $X_{ref}$, $S_{ref}$, $V_{ref}$, and $F_{ref}$ were calculated using the model described above to keep the biomass specific growth rate at a constant value. The substrate feeding rate, while building reference trajectories, was calculated as:

$$F_s = (\sigma \cdot X \cdot V + (F_b + F_{af} - F_c - F_e) \cdot S) / (S_f - S) \tag{12}$$

### 2.6. MPC Working Principle

To apply the developed MPC software for the cultivation process of certain types of microorganisms, at first it was necessary to determine the optimum reference trajectory (also termed as 'Golden Batch') for the total biomass $XV_{ref\_opt}$. This requires run-to-run fermentation optimization according to the following principles:

1.  Before starting the process, a reference trajectory for total biomass $XV_{ref.}$ is determined by using a mathematical model, the outcome of which is based on initial process conditions $X_o$, $S_o$, $V_o$, and the value of the biomass specific growth rate (which is meant to be maintained constant throughout the process). Model parameters ($Y_{xs}$, $\sigma_{max}$, $K_s$, $K_{si}$, $K_{Xmax}$) are obtained either from previously performed processes data analysis or from the scientific literature. For subsequent processes, all model parameters are determined through optimization, which is carried out based on experimental data from the previous process.
2.  After supplying data on biomass $X_t$ and/or substrate $S_t$ concentrations to the MPC software (after taking off-line samples) and based on the current volume of the culture medium $V_t$ ($V_t = V_o + V_{base} + V_{acid} + V_{antifoam} + V_{feeding} - V_{sampling} - V_{CO2} - V_{evaporation}$) in respect to the reference feeding profile, the summary total biomass trajectory $XV_{pred}$ is calculated.
3.  If the deviations from target $XV_{ref.}$ and predicted trajectories $XV_{pred}$ are above the pre-set error, the MPC software recalculates the model parameters ($\sigma_{max}$ and $Y_{xs}$). Based on the updated model parameters, the feeding rate profile $F_s$ is corrected to minimize the deviation between the target and predicted trajectories until the nearest control horizon (Figure 3).
4.  The timeframes (2–4 sampling times, can be adjusted in the SCADA interface) over which the state of the process is predicted, and the system operates in an open-loop control manner (no control action correction is implemented until new off-line data are supplied to the software), is termed as prediction horizon. The timeframe (sampling time, in our case in the range of 0.5–2 h depending on the process stage) over which the feeding profile is corrected is denoted as control horizon [30].
5.  The time when the feeding starts is determined according to the reference biomass trajectory as the time at which the predicted biomass concentration is equal to the value of the reference biomass concentration at which the feeding should start (when a certain substrate concentration is reached) (approximately 210–270 min from the beginning of the fermentation).

After performing each cultivation process, the off-line run-to-run optimization was carried out in order to update the total biomass trajectory. During off-line run-to-run optimization, the parameters of the applied mathematical model were re-estimated based on the experimental data obtained during the previous process. The second process reference trajectory was calculated using the updated model parameters and then it was applied for the next run. The described process is repeated successively throughout multiple iterations [31]. During the run-to-run optimization, model parameters such as $Y_{xs}$, $\sigma_{max}$, $K_s$, $K_{si}$, $K_{Xmax}$ were re-estimated. To determine the values of the mentioned parameters, the MATLAB function *LSQCURVEFIT* was used. This function is essentially the method of least squares.

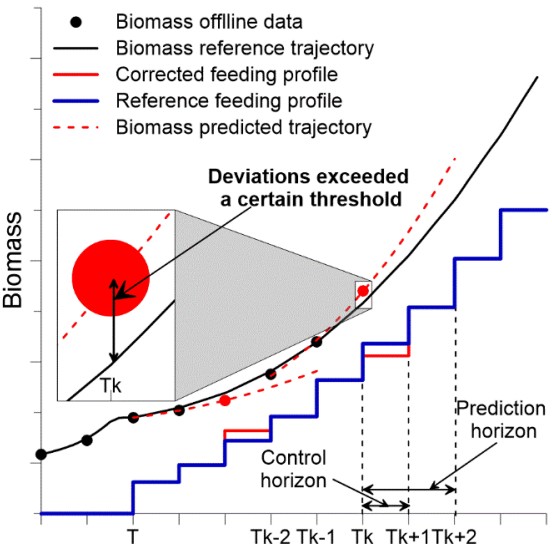

**Figure 3.** MPC operation principle [28].

Optimization of the process was considered complete when, for a given specific growth rate, the curves of reference, experimental, and predicted total biomass (XV) exhibited minimum deviations.

## 3. Results and Discussion

### 3.1. Identifying the "Golden Batch" Trajectories for E. coli BL21(DE) pET-NGF Cultivation

Validation of the developed MPC system, as well as the method for identifying model parameters and calculating target trajectories, was carried out during the cultivation of the *E. coli* strain BL21 with the biosynthesis of the recombinant protein NGF. The cultivation processes were conducted in a bioreactor equipped with a gravimetric system for controlling the supply of feeding and titrants.

To determine the 'Golden Batch' trajectory for the *E. coli* cultivation process and estimate the model parameters, the off-line run-to-run optimization method was used. To implement run-to-run optimization, a series of five cultivation experiments were carried out. In the first experiment, to model the target trajectories for the *E. coli* cultivation process (without biosynthesis of recombinant protein), the model coefficient values given in the article by O. Grigs et al. [32] for the *E. coli* BL21 strain were used. Since a different correlation equation for biomass concentration versus OD was applied in the present work, the model parameters, e.g., $Y_{xs}$, $\sigma_{max}$, $K_{Xmax}$, and $X_{crit}$ were adjusted. The values of the parameters utilized to model the target trajectories are given in Table 1. To calculate the feeding profile, the specified value of the specific growth rate $\mu_{set} = 0.3$ was used. The evaporation ($F_e$) and carbon loss ($F_c$) rates with the exhaust gases were considered zero. The experimental data on biomass growth, substrate consumption, total biomass, volume of the cultivation medium, and the feeding rate in reference to the target trajectories are shown in Figure 4.

To calculate the new (optimized) target trajectories and improve the control of the subsequent process, the following model parameters, $F_e$, $Y_{Xaf}$, $Y_{Xb}$, $Y_{OX}$, $O_m$, $t_e$, were re-estimated. The values of the parameters $Y_{xs}$ and $\sigma_{max}$ were taken as the average values of the parameters estimated online during the first process. Parameters $K_s$, $K_{si}$, $K_{Xmax}$ were left unchanged. The values of the parameters used for modeling the subsequent process are given in Table 1.

To calculate the target trajectories of the third process, model parameters $Y_{xs}$, $\sigma_{max}$, $K_s$, $K_{si}$ and $K_{Xmax}$ were once again re-estimated based on data from the two previous processes. Model parameters such as $Y_{Xb}$, $Y_{OX}$, $O_m$, and ty were also clarified. The values of the parameters are given in Table 1.

For the fourth process, the model parameters $Y_{xs}$, $\sigma_{max}$, $K_s$, $K_{si}$, $K_{Xmax}$, $Y_{Xaf}$, $Y_{Xb}$, and $t_e$ were fine-tuned based on data from the three previous processes. The values of the

parameters for modeling the following process are given in Table 1. The fifth process was carried out to test the suitability of the model and target trajectories determined after the fourth process and evaluate their adequacy as the 'Golden Batch' process.

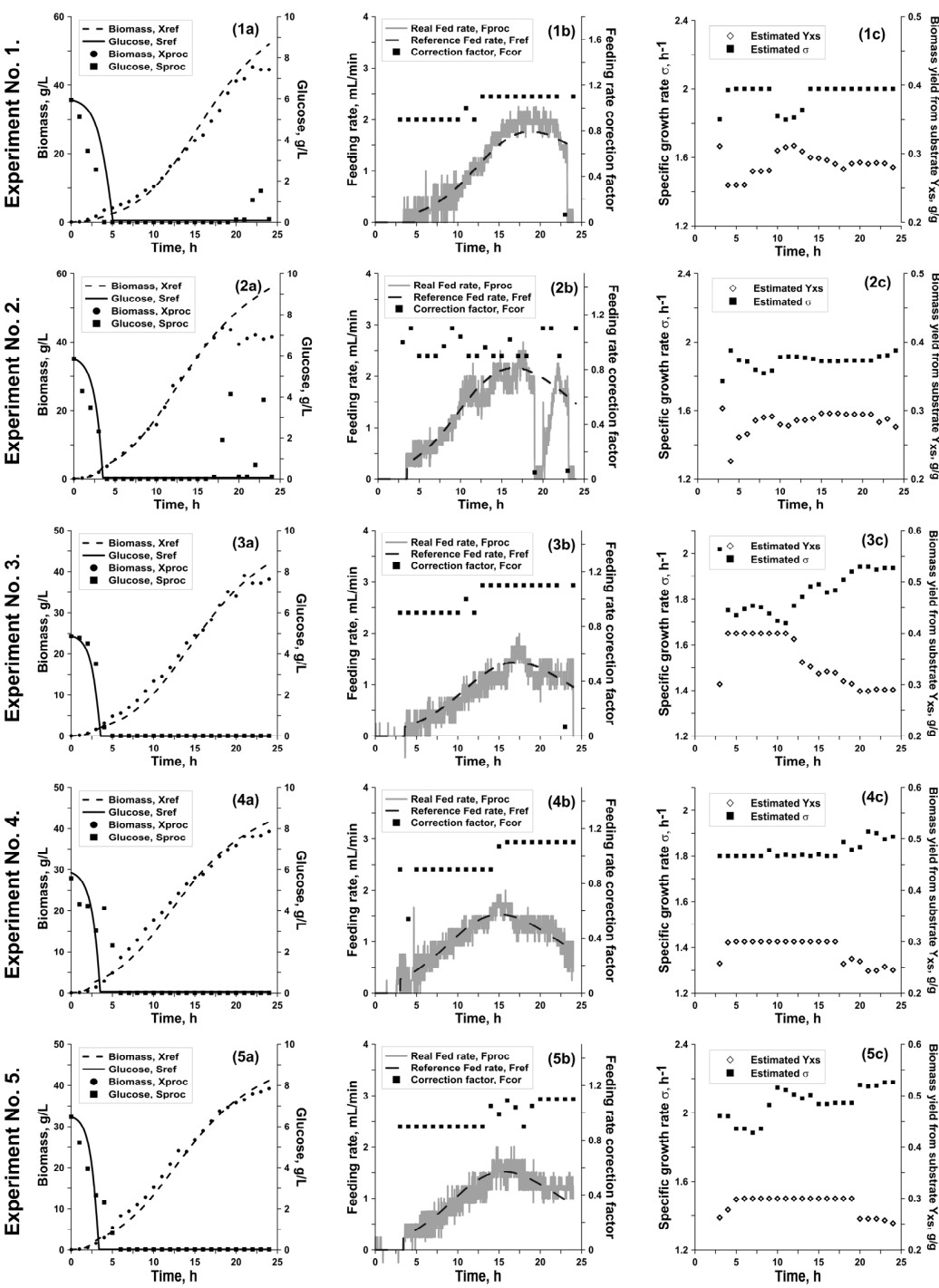

**Figure 4.** *E. coli* BL21 (DE3) NGF experimental runs employing feed rate control by MPC algorithm to estimate model parameters for biomass growth stage. (**a**) Biomass growth and glucose uptake curves. (**b**) Feeding rate and feeding rate correction factors. (**c**) Recalculations of the model parameters and ($\sigma_{max}$ and $Y_{xs}$ ).

**Table 1.** Model parameter values and process control errors for *E. coli* BL21 (DE3) NGF cultivation.

| Parameter | Experiment No. | | | | | Units |
|---|---|---|---|---|---|---|
| | **1** | **2** | **3** | **4** | **5** | |
| $Y_{xs}$ | 0.3141 | 0.285 | 0.2703 | 0.2467 | 0.2467 | $g\,g^{-1}$ |
| $\sigma_{max}$ | 1.482 | 1.963 | 2.112 | 2.3 | 2.3 | $g\,g^{-1}\,h^{-1}$ |
| $K_s$ | 0.05 | 0.05 | 0.0011 | 0.019 | 0.019 | $g\,L^{-1}$ |
| $K_{si}$ | 30 | 30 | 40.84 | 150 | 150 | $g\,L^{-1}$ |
| $K_{Xmax}$ | 65.5 | 65.5 | 50.1 | 47.3 | 47.3 | $g\,L^{-1}$ |
| $t_e$ | 3.5 | 3.5 | 2.8 | 2.8 | 2.8 | h |
| $F_e$ | 0 | $6.25 \times 10^{-5}$ | $6.25 \times 10^{-5}$ | $6.25 \times 10^{-5}$ | $6.25 \times 10^{-5}$ | $g\,h^{-1}$ |
| $Y_{Xaf}$ | $2.0 \times 10^{-4}$ | $1.5 \times 10^{-4}$ | $1.5 \times 10^{-4}$ | $2.5 \times 10^{-4}$ | $2.5 \times 10^{-4}$ | $L\cdot g^{-1}$ |
| $Y_{Xb}$ | $9.0 \times 10^{-4}$ | 0.0013 | 0.0012 | 0.002 | 0.002 | $L\cdot g^{-1}$ |
| $Y_{OX}$ | 1.185 | 0.6 | 1.185 | 1.185 | 1.185 | $g\,g^{-1}$ |
| $O_m$ | 0.0015 | 0.002 | 0.0015 | 0.0015 | 0.0015 | $g\cdot g^{-1}\cdot h^{-1}$ |
| $X_{crit}$ | 13.1 | 13.1 | 13.1 | 20 | 20 | g/L |
| $Y_{xye}$ | 0.375 | 0.375 | 0.375 | 0.325 | 0.325 | $g\,g^{-1}$ |
| $WAPE_X$ | 10.9 | 12.1 | 8.0 | 7.5 | 5.8 | % |
| $WAPE_S$ | 57.5 | 48.3 | 20.5 | 37.9 | 27.0 | % |
| $WAPE_{XV}$ | 8.73 | 13.2 | 7.9 | 7.4 | 6.7 | % |

As can be seen from Figure 4, in the first experiment the feeding start time was predicted with a noticeable error, which most likely is due to the inaccuracy of the values of the initial model parameters for calculating the target trajectories. Although, starting from the second process, after identifying such parameters as $Y_{xs}$ and $\sigma_{max}$, the feeding start time and feeding rate were determined with sufficient accuracy, keeping the desired glucose concentration at a minimum level from the very beginning of the fed-batch phase. The accumulation of the substrate and the large discrepancy between the biomass concentration and the predicted value at the end of the first process is also associated with the insufficient accuracy of the initial model parameters and, consequently, the observed prediction error (Table 1). For the first process, the weighted absolute percentage error (WAPE) of process control in the case of biomass concentration was $WAPE_X = 10.9\%$; for the substrate $WAPE_S = 57.5\%$; for the total amount of biomass $WAPE_{XV} = 8.73\%$. For the second process, the following errors were identified: $WAPE_X = 12.1\%$; $WAPE_S = 48.3\%$; $WAPE_{XV} = 13.2\%$. At the same time, Figure 4 clearly shows that the controller successfully coped with the accumulation of glucose at the end of the first and second processes, although it was not possible to achieve the desired biomass in practice. Analysis of the phosphorus content for these processes showed that its concentration after the 20th hour is close to zero, which indicates that biomass growth is limited by this component. To solve this issue in subsequent processes, $KH_2PO_4$ was added to the feeding medium so that its concentration was 6 g/L. In the third process, coefficients calculated using the data from the first and second processes were used to model the target trajectories. The control error of the third process for biomass concentration was $WAPE_X = 8.0\%$; $WAPE_S = 20.5\%$; $WAPE_{XV} = 7.9\%$.

In 'Golden Batch' experiments, e.g., the fourth and fifth cultivations, the same model parameters calculated from the experimental data of processes one, two, and three were used to build the target trajectories. As can be seen from Figure 4, the experimentally obtained trajectories of glucose and biomass concentrations have minimum deviations from the target trajectories. The minimum process control error was obtained in the fifth process—$WAPE_X = 5.8\%$; $WAPE_{XV} = 6.7\%$, which is comparable to the biomass concentration measurement error. The substrate control errors for the fourth and fifth processes were $WAPE_S = 27.0 - 29.2\%$ and, as in the third process, were associated with the batch phase, after which the glucose concentration was minimal and corresponded to the calculated trajectory.

Thus, from the stated above, we can conclude that to estimate the model parameters and obtain a 'Golden Batch' profile for the biomass cultivation processes of a given *E. coli* strain, three to five run-to-run optimization processes might be sufficient.

### 3.2. NGF Protein Biosynthesis

After identifying the "Golden Batch" trajectories for the *E. coli* BL21 biomass growth, multiple fermentation experiments were carried out with the induction of NGF recombinant protein biosynthesis after reaching a specified cell density (Figure 5). Recombinant protein synthesis was induced at DCW = 25 g/L since at this biomass concentration the number of viable cells was the highest (CFU = $1.7 \times 10^{11}$). In the first experiment with recombinant protein biosynthesis, in accordance with the calculated target trajectories it was predicted that the biomass growth parameters will not change after induction. In the second experiment, the model parameters for the recombinant protein biosynthesis stage ($Y_{xs\_p}$ and $\sigma_{max\_p}$) were adjusted using the data obtained in the first process (Table 2). The third process was a control process and was carried out to check the accuracy of the identified parameters for modeling the recombinant protein biosynthesis stage.

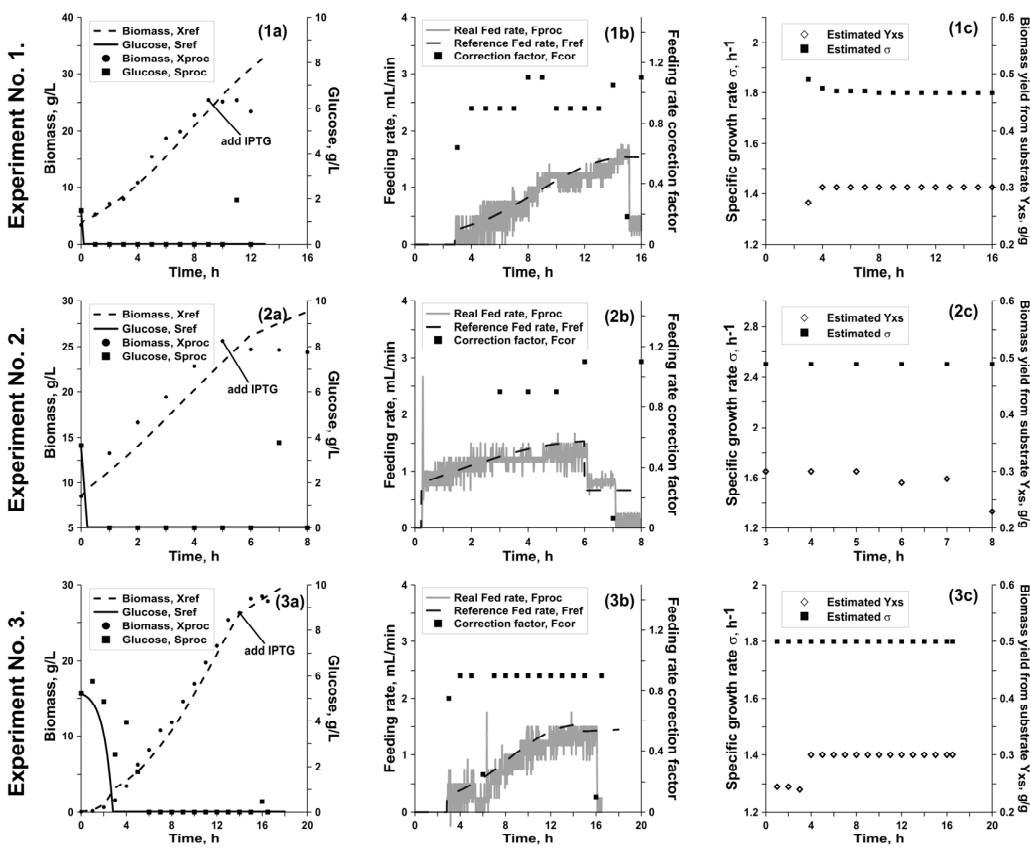

**Figure 5.** *E. coli* BL21 (DE3) experimental runs employing feed rate control by MPC algorithm to estimate model parameters for NGF biosynthesis. (**a**) Biomass growth and glucose uptake curves. (**b**) Feeding rate and feeding rate correction factors. (**c**) Recalculation of the model parameters ($\sigma_{max}$ and $Y_{xs}$)

As can be seen from Figure 5, in the first experiment, after the induction of recombinant protein biosynthesis, the biomass growth completely stopped, and the concentration of biomass began to fall due to dilution (as the feeding solution was continuously supplied). In the second hour after the induction, the accumulation of glucose in the fermentation medium was observed, indicating a change in the substrate consumption rate. The process control error for biomass concentration was $\text{WAPE}_X = 8.0\%$; for the substrate $\text{WAPE}_S = 20.5\%$; for the total biomass $\text{WAPE}_{XV} = 7.9\%$.

**Table 2.** Model parameter values and process control error for NGF biosynthesis stage.

| Parameter | Experiment No. | | | Units |
|---|---|---|---|---|
| | 1 | 2 | 3 | |
| $Y_{xs\_p}$ | 0.247 | 0.247 | 0.15 | $g\,g^{-1}$ |
| $\sigma_{max\_p}$ | 2.3 | 1.0 | 2.1 | $g\,g^{-1}\,h^{-1}$ |
| $K_{s\_p}$ | 0.019 | 0.019 | 0.019 | $g\,L^{-1}$ |
| $K_{si\_p}$ | 150 | 150 | 150 | $g\,L^{-1}$ |
| $K_{Xmax\_p}$ | 47.3 | 47.3 | 47.3 | $g\,L^{-1}$ |
| $WAPE_X$ | 13.7 | 11.5 | 7.2 | % |
| $WAPE_S$ | 56.5 | 51.4 | 45.8 | % |
| $WAPE_{XS}$ | 12.3 | 11.6 | 6.7 | % |

In the second process, after implementation of various model parameter corrections, the control error decreased. However, discrepancies between the experimental biomass growth curve and the target trajectory and the accumulation of glucose in the culture medium (up to 4 g/L) indicated the need for further fine-tuning of the model parameters. The third experiment was performed with the adjusted model parameters obtained from the first and second processes.

As shown in Figure 5(3a), the deviation of biomass growth from the target trajectory and glucose accumulation was minimal. The process control error for biomass concentration $WAPE_X = 7.2\%$; for substrate $WAPE_S = 45.8\%$; for the total amount of biomass $WAPE_{XV} = 6.7\%$ (Table 2). The observed data and the performance of the model during the third process indicate that it can be applied as a 'Golden Batch' trajectory for the biomass growth and biosynthesis of NGF by *E. coli* BL21.

*3.3. Identifying the "Golden Batch" Trajectories for E. coli BL21(DE) Cultivation and Biosynthesis of Qβ-VLPs*

The reliability (robustness) of the developed MPC-based system, as well as the method for identifying model parameters and calculating target trajectories, was tested using an *E. coli* strain that produces Qβ-VLPs bacteriophage Qβ CP, which is able to self-assemble into VLPs. The cultivation process was conducted in a bioreactor that was equipped with a volumetric control system for the supply of feeding and titrants. However, the use of a volumetric component supply system introduces additional disturbances into the process control system. This is because the system lacks feedback control to precisely regulate the number of incoming components into the bioreactor.

To identify the model parameters and determine the target trajectories of the "Golden-Batch" process, the same run-to-run fermentation optimization was used as in the previous case (bioreactor equipped with the gravimetric system). To calculate the target biomass growth trajectory, in the first process the coefficients obtained from the biosynthesis of NGF were used. In subsequent experiments, model parameters calculated from previous processes were used to calculate the target trajectories. The values of the parameters applied for modeling the subsequent process are given in Table 3. The first two experiments were carried out without inducing the synthesis of Qβ-VLPs in order to fine-tune the parameters of biomass growth. Two subsequent experiments were carried out with Qβ-VLPs induction in order to determine the model parameters for this stage of the process. The fourth process was a validation process and had the goal of assessing the adequacy of the identified model parameters for calculating the "Golden-Batch" trajectories.

Figure 6 displays the results of *E. coli* BL21 (DE) cultivations with the production of Qβ-VL bacteriophage Qβ-VLPs. As can be observed, the first experiment (where coefficients from the NGF biosynthesis were utilized to calculate the target MPC trajectories) showed a very large experimental modeling ($WAPE_X$ and $WAPE_S$ were 31 and 37.6%, respectively) and control error ($WAPE_{XS} = 25\%$). Nevertheless, in this process it was possible to obtain a rather high titer of biomass $-53.0\ g\,L^{-1}$. The accumulation of glucose was detected only in the initial phase of the process, which was due to an incorrect modeling of the batch

phase and, consequently, to the premature start of feeding. Based on the data collected in the first process, the model parameters ($Y_{xs}$, $\sigma_{max}$, $K_s$, $K_{si}$, and $K_{Xmax}$) were re-estimated and used to calculate the target trajectories of the successive process.

**Table 3.** Model parameter values and process control errors for *E. coli* BL21 (DE3) cultivation and biosynthesis of Qβ-VLPs.

| Parameter | Experiment No. | | | | Units |
|---|---|---|---|---|---|
| | 1 | 2 | 3 | 4 | |
| $Y_{xs}$ | 0.2467 | 0.33 | 0.33 | 0.33 | $g\,g^{-1}$ |
| $\sigma_{max}$ | 2.3 | 1.8 | 1.7 | 1.7 | $g\,g^{-1}\,h^{-1}$ |
| $K_s$ | 0.019 | 0.01 | 0.002 | 0.002 | $g\,L^{-1}$ |
| $K_{si}$ | 150 | 44 | 44 | 44 | $g\,L^{-1}$ |
| $K_{Xmax}$ | 47.3 | 53 | 53 | 53 | $g\,L^{-1}$ |
| $t_e$ | 2.83 | 2.83 | 3.0 | 3.2 | h |
| $Y_{xye}$ | 0.235 | 0.325 | 0.3 | 0.3 | $g\,g^{-1}\,h^{-1}$ |
| $Y_{xs\_p}$ | - | - | 0.33 | 0.238 | $g\,g^{-1}$ |
| $\sigma_{max\_p}$ | - | - | 1.7 | 1.36 | $g\,g^{-1}\,h^{-1}$ |
| $K_{s\_p}$ | - | - | 0.002 | 0.001 | $g\,L^{-1}$ |
| $K_{si\_p}$ | - | - | 44 | 89 | $g\,L^{-1}$ |
| $K_{Xmax\_p}$ | - | - | 53 | 46 | $g\,L^{-1}$ |
| $WAPE_X$ | 31.0 | 11.2 | 9.0 | 8.2 | % |
| $WAPE_S$ | 37.6 | 27.5 | 29.2 | 23.1 | % |
| $WAPE_{XS}$ | 25.0 | 7.7 | 8.6 | 12.9 | % |
| $WAPE_{Xp}$ | - | - | 7.1 | 10 | % |
| $WAPE_{Sp}$ | - | - | 98.4 | 94 | % |
| $WAPE_{XSp}$ | - | - | 15.6 | 30 | % |

In the second experiment, the errors in modeling the biomass growth and substrate uptake trajectories were significantly reduced ($WAPE_X$ and $WAPE_S$ were 11.2 and 27.5%, respectively), while the control error ($WAPE_{XS}$) reached only 7.7%. As in the first process, the calculated trajectory of the batch phase showed the largest deviation from the experimental data and the feeding had to start later than the calculated time to avoid substrate accumulation and acetate production. Since biomass growth in the batch phase is highly dependent on the content of yeast extract, coefficients such as $t_e$ and $Y_{xye}$ were re-estimated.

The third experiment of *E. coli* BL21 (DE) cultivation was carried out with the induction of Qβ-VLPs biosynthesis. For the biosynthesis stage, the coefficient of specific substrate uptake ($\sigma_{max\_p}$) was chosen to be the same as for the biomass growth stage. Induction was carried out at the 15th hour of the process with a titer of biomass equal to 28 g $L^{-1}$. As can be seen in Figure 6(3a), after induction, from the 15 to 19 h, the biomass grew according to the predicted trajectory. At the 21st hour, the biomass growth trajectory deviated greatly from the target trajectory, and glucose began to accumulate in the cultivation medium. Despite such a large deviation from the target trajectory, the control system was able to maintain the concentration of the substrate at a minimum level, preventing its further accumulation. The control error for the biosynthesis stage was $WAPE_{XS}$ = 15.6%. The process modeling error of the biomass growth stage was $WAPE_X$ = 9% for biomass and $WAPE_S$ = 29.2% for the substrate. At the same time, the control error for the biomass growth stage was $WAPE_{XS}$ = 8.6%. The SDS-PAGE analysis showed (Figure 7) that the maximum Qβ-VLPs titer was observed at the 21st hour of the process, after which the amount of Qβ-VLPs gradually decreased, which coincides with the accumulation of glucose in the medium and a sharp decrease in the feeding rate. The biomass growth modeling parameter after induction ($\sigma_{max\_p}$) was estimated (Table 3).

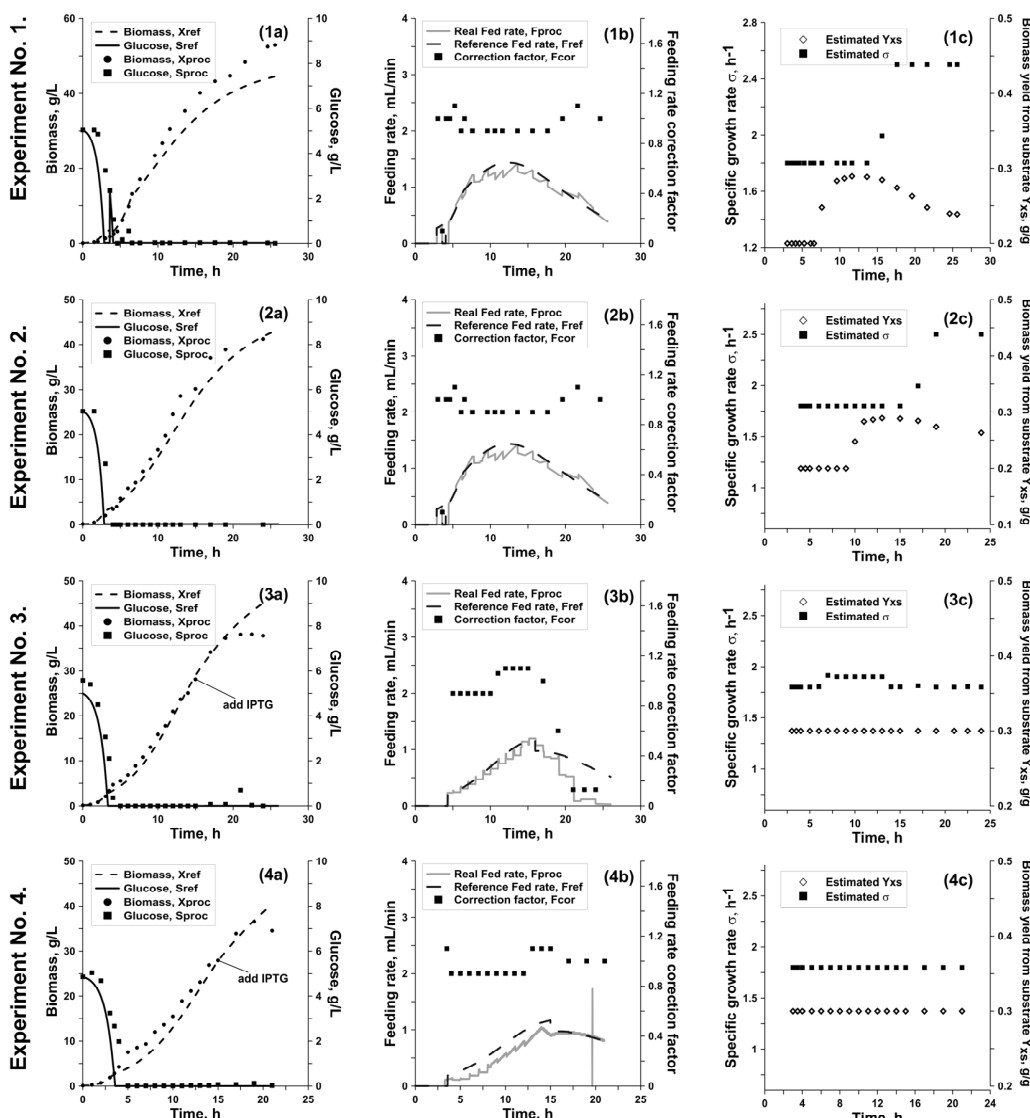

**Figure 6.** *E. coli* BL21 (DE3) NGF experimental runs employing feed rate control by MPC algorithm to estimate model parameters for biomass growth and Qβ-VLPs biosynthesis stage. (**a**) Biomass growth and glucose uptake curves. (**b**) Feeding rate, and feeding rate correction factors. (**c**) Recalculation of the model parameters ($\sigma_{max}$ and $Y_{xs}$).

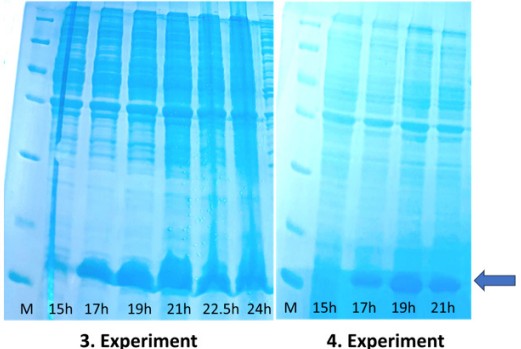

**Figure 7.** SDS-PAGE of Qβ-CP production at different time points for third and fourth experiments. Band corresponding to Qβ CP is marked with an arrow. M—molecular weight marker, bands from bottom: 14.4 kDa, 18.4 kDa, 25 kDa, 35 kDa, 45 kDa, 66.2 kDa, 116 kDa.

The fourth process, which was carried out to check the adequacy of the mathematical model coefficients, is shown in Figure 6(4a). As can be seen, despite the deviation of the experimental biomass growth curve from the target trajectory, at the initial stage of the process, the control algorithm was able to adjust the profile in such a way that the desired biomass concentration (28 g/L) was achieved by the estimated induction time. The control error for the entire process was $WAPE_{XS} = 12.4\%$ and is mainly associated with the initial stage of biomass growth.

*3.4. Discussion*

From the results obtained in the study, it can be deduced that the MPC method showed rapid adaptation to diverse cultivation systems and high resistance against internal and external disturbances. The number of experiments required to evaluate mathematical model parameters strongly depended on their initial significance, i.e., on capability to describe the system studied. Meticulous estimation of model parameters can significantly reduce the number of experiments required to optimize a model. A gradual increase in the biomass-specific growth rate ($\mu_{set}$) may be one of the solutions utilized in experiment designing to decrease the number of experiments.

It is worth mentioning that the stage of recombinant protein biosynthesis is difficult to model. Largely, this can be attributed to the fact that after the induction of recombinant protein biosynthesis, the behavior of cells and their morphology varies. That is, the estimation of biomass concentration by OD measurements becomes less accurate and may not reflect the real situation. Additionally, the stage of biosynthesis of recombinant proteins in specific cases was quite short and did not require such a careful control of the biomass, so only the substrate control was necessary.

As can be seen from the last experiments in each series, the involvement of the control algorithm (i.e., the correction of feeding profile) was minimum. Therefore, it can be concluded that the given control method is necessary only as an auxiliary tool to determine the values of the model parameters. It is sufficient to use the open loop MBC (model based control) as the main control method. This assumption will be partially true in cases where the process occurs at sufficiently low biomass-specific growth rates ($\mu_{set} < 0.25 \ \mu_{max}$) and all possible impacts are minimized. At biomass growth rates close to $\mu_{max}$, an imbalance in the system constantly arises, which requires a quick and adequate response of the control system. The system instability additionally might be referring to slightly different emergency situations that may arise in such a complex system as a bioreactor. Emergency situations vary from minor factors such as variability in the composition of the cultivation medium and the activity of the inoculum to more significant ones such as failure of control equipment (pH correction pumps or temperature control systems). The probability of such factors that affect the stability of a system increases with increasing process time. As a rule, such disturbances lead to a deceleration in the biomass growth rate and, consequently, also the rate of substrate consumption decreases. In such a situation, MPC can serve as an important assistant allowing, within reasonable limits, all kinds of system disturbances to be diminished.

## 4. Conclusions

The developed MPC method was tested in practice for the production of two different recombinant proteins (NGF and bacteriophage Qβ-VLPs) by cultivating the *E. coli* BL21 (DE3) strain. The adapted method allowed us to successfully control the biomass growth with a deviation of 6–12% from the calculated target trajectory and to maintain the minimum concentration of substrates throughout the process. Run-to-run optimization showed itself to be efficient for determining the reference process trajectories. Run-to-run optimization made it possible to obtain a "Golden Batch" profile for MPC implementation based on datasets from only 4–8 experiments. Thus, we have proven that MPC together with the run-to-run optimization method can be considered as a reliable and effective tool for controlling the cultivation of fast-growing microorganisms such as *E. coli* with the

subsequent synthesis of recombinant proteins. Furthermore, the described technology has demonstrated high adaptability, reliability, and resistance to various types of disturbances, regardless of whether they are related to the specific characteristics of the strain or the cultivation system itself.

**Author Contributions:** Conceptualization, K.D.; methodology, K.D.; software, V.G.; validation, K.D.; formal analysis, K.D.; investigation, K.D., A.S. (Arturs Suleiko), E.S., A.S. (Anastasija Suleiko), I.A. and I.P.; resources, I.P., K.T. and J.V.; data curation, K.D.; writing—original draft preparation—review and editing, K.D., A.S. (Arturs Suleiko), E.S. and K.T.; visualization, K.D.; supervision, J.V. and K.T.; project administration, K.D. All authors have read and agreed to the published version of the manuscript.

**Funding:** This research was funded by the European Regional Development Fund (ERDF) project entitled "Usage of conservative borrelia protein as vaccine target" (project No. 1.1.1.1/20/A/048).

**Institutional Review Board Statement:** Not applicable.

**Informed Consent Statement:** Not applicable.

**Data Availability Statement:** The data presented in this study are available on request from the corresponding author.

**Acknowledgments:** Bioreactors.net AS, Dzerbenes Street 27, LV-1006, Riga, Latvia.

**Conflicts of Interest:** The authors declare no conflict of interest.

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
