# Peer review of "The Application of Adaptive Model Predictive Control for Fed-Batch Escherichia coli BL21 (DE3) Cultivation and Biosynthesis of Recombinant Proteins"

_fermentation, doi:10.3390/fermentation9121015_

Round 1

Reviewer 1 Report

Comments and Suggestions for Authors

This article builds on two publications (Chem. Biochem. Eng. Q., 30 (1) 47–60 (2016) and Fermentation 2023, 9, 206) from the same laboratory. This involves Model Predictive Control of batch fermentation, and in this case control of the growth and expression by the common host organism E. coli BL21 (DE3) was demonstrated for the recombinant proteins nerve growth factor and a bacteriophage Qβ protein.

The results indicate that datasets from only a handful of experiments can be used to create reliable and flexible control algorithms for controlling fermentations before and during induction of recombinant protein expression.

There appears to be novelty in the approach, the control method appears to work well, and the article is a pleasure to read.  There are a few typographical errors (see below).

My only real concern was the lack of any comparison in the discussion of the fermentation results presented here with comparable fermentations by other researchers at the same scale - with (or without) other fermentation control systems.

Ln 24 E. coli

Methods and materials – ln 170. Here there appears a paragraph that is part of the “instruction for Authors.” and should be deleted.

Ln 194 – Bajpai – is this supposed to be a reference? I cannot find it. There is no citation reference number.

Ln 204 – the degree symbol should be used rather than a superscript zero. (37°C)

Ln 249 “foam”

Ln 316 and 323 – some grey highlighting has been left in.

Ln 365 – perhaps state that MATLAB function LSQQURVEFIT is Least Squares Fit

Ln 409, E. coli in italics

Ln 420 and other places – delete full stop in (Table 1.) Ln 476 Figure 5.(3a) Ln 543 (Figure 7.)

Figure 9 is identical to that in the author’s previous article in this journal (Fermentation 2023, 9, 206)

Figures 4 and 6 Fcor - Corection factor. Is this supposed to be a correction factor? (spelling)

References – inconsistent style – title capitals in refs 5, 25, 26, 29, 30.  Ref 28 text size; 21 Journal title in italics. Ref 6 “NPJ” should be in capitals.

Figure 9 is identical to that in the author’s previous article in this journal (Fermentation 2023, 9, 206) - will you permit that reproduction?

Author Response

Dear reviewer,

We appreciate your precious time in reviewing our paper. Your valuable comments and sharp eye for inaccuarcies in our manuscript has led to improvements in the manuscript. We hope the manuscript after more careful revisions meets your standards. We have corrected all shortcomings and errors pointed out by you. Additionally, we have tried to strengthen the paper supplementing introduction section to outline the motivation for the study and the specific challenges addressed. We now ensure more consistent terminology throughout the manuscript. We have carried out minor redaction of the manuscripts text, which improve the understanding of the presented material without changing its essence. We took into account your concerns about the Figure we did publish already in one of our previous articles and added the corresponding reference (Bolmanis et al., 2023). The list of references was supplemented with one more literature source (Bajpai R., 1987), which was missing in the initial manuscript.

All changes are marked in the text with "Track Changes". We are returning the corrected manuscript for the second review stage.

Thank you very much for your consideration.

Sincerely yours,

Konstantins Dubencovs

Reviewer 2 Report

Comments and Suggestions for Authors

Dear Authors,

I hope this message finds you well. I am writing to express my appreciation for the effort you have put into the manuscript titled "The Application of Adaptive Model Predictive Control for Fed-Batch Escherichia coli BL21 (DE3) Cultivation and Biosynthesis of Recombinant Proteins." Your work demonstrates a thorough investigation into applying model predictive control (MPC) for optimizing the cultivation and biosynthesis of recombinant proteins in Escherichia coli BL21 (DE3). I commend your systematic approach, detailed experimental setup, and the inclusion of valuable data analysis.

I have provided a comprehensive review of your manuscript, highlighting its strengths and suggesting areas for improvement. The manuscript's logical flow, comparative analysis, and transparency regarding limitations contribute to its overall credibility. However, I have also identified areas where additional clarity and context could enhance the manuscript's impact.

I encourage you to consider the suggestions in the review, especially in improving the introduction and motivation, ensuring consistent terminology, and discussing the generalizability of results. Additionally, providing more details on model validation and addressing the long-term stability of the MPC method would strengthen the manuscript.

Your work holds great promise, and I believe addressing these suggestions will contribute significantly to the manuscript's overall quality. I appreciate your dedication to advancing research in this field, and I look forward to seeing the continued progress of your work.

Author Response

Dear reviewer,

We appreciate your precious time in reviewing our paper. We would like to express gratitude for looking through our work and putting a noticeable effort into helping us elevate the quality of our article. All valuable comments, outlined shortcomings, and suggestions were taken into account by our team and have been implemented in the submitted version of the manuscript. We hope that the manuscript after improvements, supplemented information and careful revisions meets your standards and has strengthen the paper. We now ensure more consistent terminology throughout the manuscript, also, we have supplemented the introduction section to outline the motivation for the study and the specific challenges addressed. We have carried out minor redaction of the manuscripts text, which improve the understanding of the presented material without changing its essence. The list of references was supplemented with two additional literature sources (Bajpai R., 1987, and Bolmanis et al., 2023), which were missing in the initial manuscript.

All changes are marked in the text with "Track Changes". We are returning the corrected manuscript for the second review stage.

Thank you very much for your consideration.

Sincerely yours,

Konstantins Dubencovs
